# A Novel Airborne Molecular Contaminants Sensor Based on Sagnac Microfiber Structure

**DOI:** 10.3390/s22041520

**Published:** 2022-02-16

**Authors:** Guorui Zhou, Siheng Xiang, Hui You, Chunling Li, Longfei Niu, Yilan Jiang, Xinxiang Miao, Xiufang Xie

**Affiliations:** 1Laser Fusion Research Center, China Academy of Engineering Physics, Mianyang 621900, China; zhougr@caep.cn (G.Z.); xiangsiheng1230@163.com (S.X.); yhworkyx@163.com (H.Y.); niulf@caep.cn (L.N.); jiangyilan1023@163.com (Y.J.); miaoxx@caep.cn (X.M.); 2Institute of Applied Electronics, China Academy of Engineering Physics, Mianyang 621900, China; odiaha@sina.com

**Keywords:** airborne molecular contaminants (AMCs), Sagnac microfiber loop, high power lasers (HPLs)

## Abstract

The impact of airborne molecular contaminants (AMCs) on the lifetime of fused silica UV optics in high power lasers (HPLs) is a critical issue. In this work, we demonstrated the on-line monitoring method of AMCs concentration based on the Sagnac microfiber structure. In the experiment, a Sagnac microfiber loop with mesoporous silica coating was fabricated by the microheater brushing technique and dip coating. The physical absorption of AMCs in the mesoporous coating results in modification of the surrounding refractive index (RI). By monitoring the spectral shift in the wavelength domain, the proposed structure can operate as an AMCs concentration sensor. The sensitivity of the AMCs sensor can achieve 0.11 nm (mg/m^3^). By evaluating the gas discharge characteristic of four different low volatilization greases in a coarse vacuum environment, we demonstrated the feasibility of the proposed sensors. The use of these sensors was shown to be very promising for meeting the requirements of detecting trace amounts of contaminants.

## 1. Introduction

Airborne molecular contaminants (AMCs) from material outgassing and shedding continue to be a challenging problem for optics components in high power lasers (HPLs). In order to avoid breakdown of air near the focal region on propagation of intense laser radiation, the optics components used as spatial filters/optical replays were mounted on in a vacuum environment. Compared to the atmosphere, AMCs are more easily produced by slow outgassing of material present in a vacuum environment, such as vacuum pump oil, cables, residual organic compounds of optics and mechanical component sub-surface. The AMCs result in the reduction in the laser-induced damage threshold (LIDT) of optics components [1,2,3,4] and the increase in reflectivity of the sol-gel porous silica antireflection coating [5]. Thus, online monitoring of the concentration of AMCs in a vacuum environment is a critical issue for maintaining the function of the optics components under “Safety Red line”. Based on our research findings, the measured AMCs concentration ranged from 0.253 mg/m^3^ to 46.228 mg/m^3^ by using the gas chromatograph-mass spectrometer (GC-MS) technique during the operation of HPLs. On the other hand, other detection methods for AMCs are limited in harsh scenarios, especially in a vacuum or intense laser radiation environment [6,7]. Therefore, it is different to provide a more appropriate way for in-situ measurement of trace amounts of AMC in HPLs.

From the viewpoint of the optical fiber sensors structural design, sensors with thinner diameter structure have successfully achieved higher sensitivity [8]. Compared with the standard optical fiber for optical communication systems, optical microfibers (OMFs) have diameters ranging from hundreds of nanometers to tens of micrometers. Light guided along such OMFs leaves a large fraction of guided field outside the OMFs as evanescent waves to penetrate the environment, which allows the direct interaction between light and its surroundings [9,10,11]. Due to its thin diameter, the OMFs provide other commendable features, such as outstanding mechanical flexibility, tight optical confinement and low optical loss [12]. There are several major types of OMFs sensors that have been developed recently, which include Mach–Zehnder OMFs interferometric sensors [13,14,15,16,17], OMFs Bragg/long period grating (FBG)-based sensors [18,19,20], OMFs coupler sensors [21,22,23,24], Sagnac OMFs interferometric sensors [25,26,27,28,29], and OMFs multimode interferometric sensors [30,31,32]. The various types of sensors, depending on their unique structures, possess different sensing performances, in terms of sensitivity and temperature stability. Due to its truly path-matched interference mode, the Sagnac interferometric sensor has shown its superior temperature stability and high sensitivity, which renders it suitable, particularly for AMCs sensing.

In this paper, we proposed and investigated a novel AMCs sensor structure consisting of a Sagnac OMFs loop and a mesoporous silica coating on the OMFs surface. The experimental verification based on the proposed configuration was provided, along with the theoretical analysis. The AMCs sensors based on the Sagnac OMFs loop were fabricated by the heating brushing technique and by dip coating after synthesizing the sol-gel solutions. Verifying the proposed AMCs sensors indicated that they would have high sensitivity. In addition, to achieve the in-situ monitoring of typical AMCs, such as dibutyl phthalate (DBP), a gas discharge characteristic qualitative evaluation of various greases in a vacuum environment was done to verify the technical feasibility of the AMCs sensors. Furthermore, the sensors provided supporting measures of optics components operation, which is suitable for potential application in biological and chemical detection. Compared with our previous work [33], this work has made progress on novel structure and theoretical analysis, as well as the measurement method and technical feasibility verification. Firstly, taking the advantage of energy transfer in the coupling region, we reused the region as the effective sensing region; the device with the Sagnac OMFs loop possesses a more compact structure than a system containing an OMFs coupler. Secondly, the Sagnac OMFs loop acted as a reflector that reflects two beams from the OMFs coupler and that enhances interaction between the evanescent field and the external environment when the two beams re-enter OMFs and interfere. The sensors based on the Sagnac OMFs loop can achieve higher sensitivity. Therefore, we have carried out theoretical analysis and formular derivation for the OMFs sensor based on Sagnac interference. Finally, during the experiment, we optimized the experiment setup to further eliminate the measurement errors and to enhance the accuracy of the sensor performance measurement. Simultaneously, owing to its unique structure, the Sagnac OMFs loop is characterized by more symbolic wavelength dips, better recognizability in the measurement process, and better usability. In addition, its technical feasibility has been carried out in this work.

## 2. Sensor Configuration and Principle

Figure 1 shows the schematic configuration of the proposed AMCs sensor. It consisted of a Sagnac OMFs loop and mesoporous silica coating on the OMFs surface; the Sagnac OMFs loop contained an OMFs coupler and a Sagnac loop. The method of fabricating the Sagnac OMFs loop has already been studied in a previous report [33]. As shown in Figure 1, light injected into port P_0_ could be coupled and split into clockwise and counterclockwise beams by the OMFs coupler, which propagates along the Sagnac loop; the interference was given by the recombination of the counter-propagating beams at the coupler. The principle of the sensor is that the beating between the lowest order even supermode and odd supermode provides energy transfer between two OMFs; wavelength dips in spectral region can be obtained as a result of the optical path difference between the two supermodes accumulated along the coupling region [34].

In contrast, weakly fused OMFs couplers own a stronger dependency on ambient RIs than the strongly fused ones and are more suitable to work as sensing units [35]. Therefore, only weakly fused OMFs couplers were studied and fabricated in this thesis. Both OMFs roughly maintained original cross-section geometry throughout the entire fabrication process due to the larger aspect ratio, defined as the ratio between the total width and the height of the cross section in the uniform region (generally, larger than 1.8) [36]. Thus, it is reasonable as an acceptable simplification that two identical OMFs with a uniform shape and RI profile are fused together. Due to weakly fused OMFs in the uniform region, weak coupling theory was employed for the Sagnac OMFs loop sensing structure. The coupling coefficient *C* of the whole coupling region can be approximately denoted as [37]:(1)$$C\left(\lambda \right)=\frac{\pi \sqrt{n_{1}^{2}-n_{2}^{2}}}{2an_{1}}e^{-2.3026\left(A+B\tau +C\tau^{2}\right)}$$
$$A=a_{1}+a_{2}V+a_{3}V^{2}\;a_{1}=2.2926\;a_{2}=-1.591\;a_{3}=-0.1668\;\;B=b_{1}+b_{2}V+b_{3}V^{2}\;b_{1}=-0.3374\;b_{2}=0.5321\;b_{3}=-0.0066\;C=c_{1}+c_{2}V+c_{3}V^{2}\;c_{1}=-0.0076\;c_{2}=-0.0028\;c_{3}=0.0004$$
where *λ* is the wavelength of the incident light, *n*_1_ and *n*_2_ refer to the RIs of Sagnac OMFs loop and the surrounding medium, *a* is the radius of OMF in the waist region, *τ* = *d*/*a* is the aspect ratio of the cross section of the OMFs coupler, *d* is the distance between the axis of the fused OMFs and *V* is the normalized frequency where *V* = [(2*πa*)/*λ*]·(*n*_1_^2^ − *n*_2_^2^)^1/2^. The coefficients *a*_1_, *a*_2_…*c* are functions of the shape of the OMFs couplers. One can approximate the modal field distribution of the fundamental mode by the Gaussian-exponential-Hankel function, whose parameters are determined by using a variable technique and the coupled-mode theory [37]. Based on these, we have found that, even for a step-index OMFs coupler, an empirical relation between the coupling coefficient *C* and *V* can again be described; the values of the coefficients for the OMFs coupler are given. When incident light of power enters the *P*_0_ input port, the power at the *P*_1_ output port can be expressed as [38]:(2)$$P_{1}=\frac{1}{2}\left[1+cos^{2}\left(2CL\right)\right]$$
where *L* is the coupling length of the Sagnac OMFs loop. According to Equation (1), the coupling coefficient *C* is related to the RI of the surrounding medium, the incident wavelength *λ* and the coupler radial size 2*a*. Considering effective RI modification of mesoporous coating after the physical absorption of AMCs, the wavelength of the dip and intensity in the output interference spectrum varied as AMCs concentration in external environments changed when the silica mesoporous coating was used as the surrounding medium of the Sagnac OMFs loop. Therefore, the measurement of AMCs concentration could be realized by detecting the wavelength shift of the dip and variation in the spectral domain. According to Equations (1) and (2), the sensitivity of the wavelength shift of the dip to the RI change of the surrounding medium can be derived as
(3)$$\frac{\partial \lambda }{\partial n_{2}}=-\frac{\partial \left(2CL\right)/\partial n_{2}}{\partial \left(2CL\right)/\partial \lambda }=\frac{4\pi^{2}a^{2}n_{2}}{V^{2}}\left(\frac{1}{DV\lambda }-1\right)$$
$$D=a_{2}+2a_{3}V+b_{2}+2b_{3}V+c_{2}+2c_{3}V$$

Here, *V* is the normalized frequency, *a* is the radius of OMF in the waist region, and there is a linear relationship between *D* and *V*. For the Sagnac OMFs loop structure, *V* > 1. Therefore, it is easy to obtain $\frac{\partial \lambda }{\partial n_{2}}<0$, which indicates that the wavelength of the dips blueshifts as the RI of the surrounding medium *n*_2_ increases. Besides, according to the results of numerical simulation for single mode fiber [12], there are various modes existing in the coupling region when *V* > 1 and the diameter of OMFs is greater than 800 nm, even appearing in several high-order modes (e.g., HE31). After preliminary calculation, when the diameter of the waist region of Sagnac OMFs loop was 3.0 μm and the incident wavelength was 1550 nm, there were more than 10 modes in the coupling region.

## 3. Experiment

### 3.1. Silica Sensitive Coating Solution Preparation

Here, we employed a facile one-step solution-based synthetic route to produce silica sensitive coating material using controlled hydrolysis of tetraethyl orthosilicate (TEOS) via the Stober method. Firstly, base-catalyzed silica solution was fabricated by stirring the mixture consisting of EtOH, TEOS, H_2_O and NH_3_·H_2_O for 2 h at 30° and aged at 25° for 7 days; the molar ratio of EtOH, TEOS, H_2_O and NH_3_·H_2_O was 1:3.25:37.6:0.17. After refluxing for 24 h to remove ammonia and filtering through a 0.22 μm PVDF filter, the final concentration of SiO_2_ in the solution was 3% by weight.

Figure 2 illustrates an SEM image of fabricated SiO_2_ nanoparticles utilizing the Stober method. As shown in the SEM image, the shapes of silica nanoparticles with a diameter of 20.0 nm were not very regular, and the mesoporous coating formed on the glass slide substrate by dense packing and or a van der Waals attraction. In addition, porosity increased due to abundant voids of coatings [39].

### 3.2. Fabrication of Sagnac OMFs Loop Sensing Unit and Experimental Setup

To fabricate the Sagnac OMFs loop, a piece of standard telecom single-mode fibers (SMF28, Corning) was firstly bent, and the two free ends were twisted together carefully to form a Sagnac loop. In order to ensure that the fibers remained in contact commendably and to prevent fibers fracturing during fabrication, the distance between two adjacent fiber knots should be kept from 4 mm to 6 mm. The twisted region was entirely fused and tapered using the microheater brushing technology. The tapering process was executed on the homemade taper drawing system, which consists of three translation stages driven by linear motors with 150 nm of positioning accuracy. In order to obtain the Sagnac OMFs loop with the required shape and diameter in the waist region, the fabrication process of a low-loss Sagnac OMFs loop was accurately controlled by a computer program.

Considering their outstanding compatibility and stability, aluminum alloy and polydimethylsiloxane (PDMS) were employed to package the proposed Sagnac OMFs loop in the aluminum alloy holder. Based on the dip coating method, the packaged Sagnac OMFs loop was dipped into prepared SiO_2_ solution mentioned above for at least 1 min, and permeated totally. The silica solution concentration and pulling rate can control the thickness of the SiO_2_ coating on the surface of the Sagnac OMFs loop. In the coating fabrication process, the fabrication setup moved upwards at the pulling speed of 200 mm/min; in order to enhance the mechanical properties of the coating, the proposed Sagnac OMFs loop with mesoporous coating was exposed in hexamethyldisilazane (HDMS) vapor in a clean room (ISO 5) by converting the remaining hydroxyl groups to trimetrylsiloxy groups. The geometry and surface morphology of the proposed sensing unit was examined under a scanning electron microscope (SEM). The SEM image of a short section of the sensing unit was shown in Figure 3a. Both fused OMFs that roughly maintained their original geometry further confirmed the weakly fused status of the fabricated Sagnac loop. Each OMFs possessed a diameter of ~2.87 μm and showed very outstanding diameter uniformity in the coupling region. The length of the waist was about 7 mm. Due to the partial overlapping of two OMFs, it could not see the complete shape of the other OMFs. Figure 3b shows a mesoporous silica coating on the surface of the fiber. An SEM image of the cross section of the proposed Sagnac OMFs loop sensing unit is illustrated in Figure 3c. Figure 3c shows the fabricated mesoporous silica coating on the surface of the Sagnac OMFs loop; the thickness of the coating was ~88.2 nm.

Due to poor mechanical stability, aluminum alloy holders with a micro-channel for washing the sensing unit were machined to fix the sensing unit. The holders did not produce any contaminants to eliminate measurement error, especially after ultrasonic cleaning. The experimental setup for the characterization of the AMCs sensor is described in Figure 3d. Incident light emitted from a semiconductor laser diode (SLD, Thorlabs Inc., Newton, NJ, USA, S5FC1550P-A2) with a center wavelength of 1550 nm was injected into the input port P_0_; the output light from P_1_ was detected by an optical spectrum analyzer with a resolution of 0.02 nm (AQ6370, Yokogawa Co. Ltd., Tokyo, Japan). The sensing units were intentionally contaminated by outgassing DBP; the aim of this experiment was to simulate an optics contamination phenomenon in order to determine contamination kinetics and the proposed AMCs sensors sensitivity. DBP was used as a model AMC for the present experiments. Owing to extensive use as plasticizers, DBP and its less volatile homologue dioctalphthalate (DOP) are very common during the building of HPLs. Moreover, DBP was also used in this experiment because of its convenient transport properties and easy diffusion in a common environment. The material was outgassed at a heating temperature that ranged from 30 °C to 70 °C; the clean compressed air (ISO 3) charged with contaminants was directed towards the proposed sensors at a flow rate of 2 L/min in order to help condensation of organics. A weighing bottle filled with DBP was heated by a magnetic stirrer with precise control of temperature (IKA RET/T). The magnetic stirrer and the proposed sensors were put in two chambers with a gas inlet and a gas outlet, respectively. We delivered the clean compressed air into the heating chamber; the mixture of the clean air and AMCs led into the measurement chamber further affected the properties of the sensors. In the measurement process, at a heating temperature, we found that the volatilization rate of DBP was basically constant (except for the first two minutes). Thus, it is reasonable to believe that the concentration of AMCs in the mixture was basically constant at the heating temperature. When a certain concentration of mixed gas gradually filled the measurement chamber, the excess gas would also be discharged through the outlet. Finally, at the heating temperature, a constant AMCs concentration would be found in the measurement chamber. Therefore, it can be seen that the heating temperature corresponded to a specific AMCs concentration.

It is important to investigate temperature characteristics of the proposed sensor in its application environment, in which they exhibit inaccuracies under special conditions, such as high-humidity and an environment with strong electromagnetic interference. The proposed sensor was placed in a thermostat with precise control of temperature and a commercial temperature sensor (Rotronic HC2A-S). As shown in Figure 3e, as the *T* increased from 30 °C to 65 °C, the wavelength dip exhibited a redshift and a decrease in intensity due to the combined effects of the thermo-optics and the thermal expansion in the tapered waist region. Here, since the diameter of OMFs is very thin, the fiber core can be neglected [40]. The change of fiber cladding owing to thermo-optic effect in the waist region plays a key role. Thus, an increase in *T* mainly results in an increase in the RI of the fiber cladding [41]. Figure 3f illustrates the response of the wavelength and intensity of the dip as ambient temperature *T* increased. It suggests polynomial fitting with an R-square of 0.9975 and 0.95745 for wavelength and intensity of the dip, respectively. Simultaneously, the fitting coefficients of wavelength and intensity of the dip were 0.03077, −0.00119 and −0.09215, −0.00332, respectively. When the ambient temperature increased from 30 °C to 65 °C, the wavelength dip blueshifted by 2.79 nm. Moreover, the transmission loss of the dip increased by 13.02 dB. The wavelength shift and intensity of the dip possess a complex polynomial relationship with *T* change; *T* sensitivities of approximately −79.7 pm/°C and −0.372 dB/°C, respectively, were achieved. When ambient temperature varied, the change of the phase difference induced by the combined effects of the thermo-optics and the thermal expansion in the tapered waist region was dominant in the process. In addition, the phase difference depended on birefringent coefficient B and the length of the twist area L. The free spectral range (FSR) of the transmission spectrum would become narrower and narrower, as the number of twist turns increased. In this situation, considering that the birefringence coefficient B mainly depends on the shape of the twist area’s cross-section and the temperature almost has no influence on B, ∂B/∂T could be regarded as zero. As a result, we were able to eliminate or solve the cross-sensitivity between AMCs and temperature by adjusting the number of twist turns in the proposed sensors preparation; theory derivation and experimental demonstration can be seen in detail in Reference [42].

### 3.3. Results and Discussion

In order to accurately obtain the concentration of DBP at different heating temperature, weight difference before and after heating was measured with micro-analytical balance (Mettler Toledo XP56) five times, to obtain the average value of weight difference. Considering the flow rate of the loading gas, the concentration of AMCs could be obtained. In order to verify the response time of the proposed sensors, the duration time of heating DBP was set to 20 min, and then the gas path and chambers were flushed by the clean air to avoid the impact of the condensation of AMCs on experimental results. The stability/repeatability of the proposed sensor mainly depended on the profile of OMFs and the influence of the background environmental cleanliness. In the fabrication process of Sagnac Loop OMFs, the evolution of the transmission spectrum of the tapered samples has been adopted to obtain required samples accurately. Meanwhile, we carried out high-spraying and ultrasonic cleaning of the chamber and its affiliated pipelines to avoid the influence of the background environmental contaminants, as mentioned above. Figure 4 illustrates transmission spectra evolution of the proposed sensors with 3.0 μm diameter at 30 °C, 50 °C and 70 °C. At heating temperatures of 30 °C, 50 °C and 70 °C, the wavelength dips shift approached shorter wavelength as the AMCs concentration increased. Simultaneously, it is noted that the higher the heating temperature, the larger the wavelength dip shift. Figure 4 also shows that the theoretical analysis has a good match with the experimental results. The experimental results shown in Figure 4 can be explained by RI variation of the silica sensitive coating and the polarization state of the input light in the waist region. The adsorption number of AMCs in the mesoporous coating increased with the increase in the concentration of DBP inside the measurement chamber, which resulted in the variation of coupling coefficients of the Sagnac loop. This caused the wavelength of the dip blueshifts on transmission spectrum. One factor that must be accounted for is that the polarization state of the input light changed as the adsorption number of AMCs increased in the sensing process. This would affect the spectral response; the overall spectrum was modified such that some peaks were considerably attenuated, even when distorted at 70 °C.

Figure 5a shows the wavelength dips shift throughout the experiment. The wavelength dip shifts of 5.4 nm, 17.9 nm and 24.8 nm correspond to the heating temperature of 30 °C, 50 °C and 70 °C, respectively. When the AMCs concentration was increasing, it appeared as a sharp decline and then became steady quickly, revealing good stability. As shown in Figure 5a, we could observe that the proposed sensors showed almost the same response time (about 14 min) at different heating temperatures, due to the same thickness of the mesoporous silica coating of the sensors. At the same concentration of AMCs, for the sensitive coating with the same thickness, it reached equilibrium between adsorption and desorption of AMCs in the same amount of time. In order to further demonstrate the experimental results, we prepared the AMCs sensors with 2.5 μm diameter at the same pulling speed. The results indicated that the response time of the proposed sensors with 2.5 μm diameter was reduced to 7.5 min. The thinner sensitive coating of the proposed sensors with 2.5 μm diameter is a major reason why the sensors own the shorter response time; compared with the thicker coating, the thinner coating more easily reached equilibrium between adsorption and desorption of AMCs under the same condition.

Figure 5b plots the measured wavelength dip change as a function of the concentration of AMCs at different heating temperatures. In the experiment, the heating temperature varied from 30 °C to 70 °C at intervals of 10 °C. In order to achieve equilibrium between adsorption and desorption of AMCs, the duration time of heating was set at 20 min and the spectral response at each temperature was recorded. It showed a polynomial fitting with an R-square of 0.95052; the fitting coefficients were 0.18558 and −2.7657 × 10^−4^. The wavelength dip shifted from 5.4 nm to 24.8 nm as the concentration increased from 0 to 213 mg/m^3^. The average sensitivity was ~0.11 nm/(mg/m^3^). Compared with the previous microfiber structure, the mesoporous silica coating allows for a stronger evanescent-field interaction between AMCs and the sensing waveguide, due to its high porosity and limited thickness.

### 3.4. The Technical Characteristic of the Proposed Sensors

The technical characteristics of the proposed sensors are closely interrelated to adsorption of the mesoporous sensitive coating, whose porosity directly determines the number of AMCs adsorbed in the sensing process. Consequently, it is reasonable to assume that a single OMF sensor with the same parameters possesses the same technical characteristics as the Sagnac loop sensor discussed here. In order to obtain characteristics of the Sagnac loop sensor, we prepared the single OMFs sensors with the same parameters. In the experiment, a piece of standard telecom single-mode fibers (SMF, Corning) was fixed on the homemade taper drawing system and tapered into OMF with a 3.0 μm diameter. Although we could accurately control the diameter and shape of OMFs through the taper drawing system, the prepared samples were be placed under the optical microscope to check whether its diameter met the requirement, so as to reduce the error induced by different diameters. The coating preparation method and parameters of single OMFs were completely consistent with those of Sagnac Loop sensors.

The physical adsorption of AMCs in the mesoporous coating of OMFs resulted in modification of the surrounding of RI; the additional loss was produced due to the absorption of the evanescent field. In the experiment, the additional loss caused by the adherence of AMCs was recorded in real time. A DFB laser at 1310nm center wavelength was divided into two beams by a light coupler, one of which was used as the signal light and the other as the reference light to eliminate the error caused by laser power fluctuation. Furthermore, in order to improve measurement accuracy, lock-in amplification technology was adopted in the process of optical signal detection and processing. The photodetector, A/D converter and DSP mode-locked amplifier were integrated into a complete measurement system with a dynamic measurement range of about 25 dB; our own software was developed to process and store the monitoring data. We employed the experimental setup shown in Figure 3d to evaluate technical characteristics of the OMFs sensors. The heating temperature was gradually increased to reach the maximum measurement concentration of the OMFs sensors. When the heating temperature of AMCs was 98 °C, the response of the OMFs sensor was shown in Figure 6. Once the additional loss achieved the maximum value of the measurement system, heating stopped immediately and the AMCs concentration of 438 mg/m^3^ was obtained through the weighing method mentioned above. One can see that the sensor possessed a maximum concentration of 438 mg/m^3^ and a response time of 17 min, which was slightly larger than the experimental results of the Sagnac Loop sensors discussed above. Thus, it can be considered that the fabricated Sagnac Loop sensors also possessed a maximum concentration of 438 mg/m^3^. However, we employed the heating method to realize the volatilization of AMCs; the minimum concentration of the proposed sensors could not be achieved due to control accuracy and condensation of AMCs in the gas path. We could obtain the sensing system resolution of 0.18 mg/m^3^, resulting from an average sensitivity of 0.11 nm/(mg/m^3^) and the OSA with the resolution of 0.02 nm used in the experiment.

As mentioned above, compared with other types of sensors, the Sagnac Loop sensor possesses more symbolic wavelength dips and better recognizability, especially for the sensor with a diameter of 3.0 μm. According to [43], the interval between the two adjacent characteristic wavelength dips was 70 nm, which was larger than the shift of wavelength dips induced by the maximum concentration of AMCs (about 48 nm). Therefore, based on our actual detection requirements, the concentration of AMCs in HPLs was less than 200 mg/m^3^; the dynamic range of the sensing system in the spectrum domain could fully meet the detection requirements of maximum concentration.

### 3.5. The Feasibility of the Proposed Sensors

In order to validate the feasibility of the proposed sensors, the proposed sensor with 3.0 μm diameter was employed to qualitatively evaluate the gas discharge characteristic of four different low volatilization greases in a coarse vacuum environment. In the experiment, the measured grease samples were put into a vacuum chamber, connected to a dry vacuum pump (Agilent TriScroll 300) with high pumping speed. Before the experiment, in order to avoid the influence of the background environmental cleanliness, we carried out high-spraying and ultrasonic cleaning of the chamber and its affiliated pipelines. First, we utilized the vacuum pump to suction out the air; the negative pressure in the vacuum chamber was maintained at ~20 Pa. The negative gauge pressure was measured using a vacuum gauge (Televac CC-10) in the vacuum chamber. Then, via a switch air hole, air was let into the vacuum chamber to achieve the fine-tuning of vacuum degree; the spectra were recorded simultaneously. The duration time of the pumping process was 20 min. Transmission spectra comparison of grease samples 00#, 02#, 82# and 83# before and after evaluation are shown in Figure 7. We found different wavelength dip shifts for grease samples 00#, 02#, 82# and 83#, as well as a certain degree of spectra distortion due to transmission fiber deformation in the vacuum environment.

Figure 8 illustrates the wavelength dips shift of grease samples 00#, 02#, 82# and 83# at a coarse vacuum of ~20 Pa. For grease sample 00#, the wavelength dip shift was about 4 nm, less than the dip shift of other grease samples. The wavelength dip shift of sample 82# was the largest, about 12 nm. The dips shift of samples 02# and 83# were 5.4 nm and 7.5 nm, respectively, between the dips shift of samples 00# and 82#. From the experimental results, it can be clearly seen that grease sample 00#, with less volatilization, was more suitable for the vacuum environment of HPLs, while grease sample 82# was the opposite. Moreover, the results demonstrate the feasibility of the proposed sensors.

## 4. Conclusions

An AMCs sensor based on the Sagnac microfiber structure for molecular contaminants concentration on-line measurement is presented. The device is fabricated by the microheater brushing technique and dip coating. For the DBP measurement as a model AMCs, the average sensitivity of the sensors was 0.11 nm/(mg/m^3^). Gas discharge characteristic qualitative evaluation of several greases in the vacuum environment was done to verify the technical feasibility of the AMCs sensors. The sensing performance of the proposed sensor can be further improved through the parameter optimization of the structure. The sensor is inexpensive and easy to fabricate; thus, it has shown great potential for in-situ measurement of AMCs in HPLs.

## Figures and Tables

**Figure 1 sensors-22-01520-f001:**
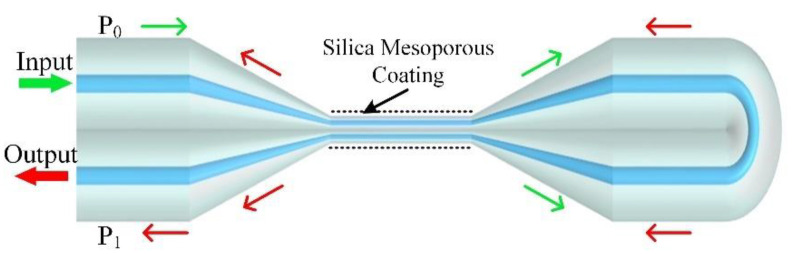
Schematic diagram of the AMCs sensor that was fabricated by employing the microheater brushing technique and dip coating.

**Figure 2 sensors-22-01520-f002:**
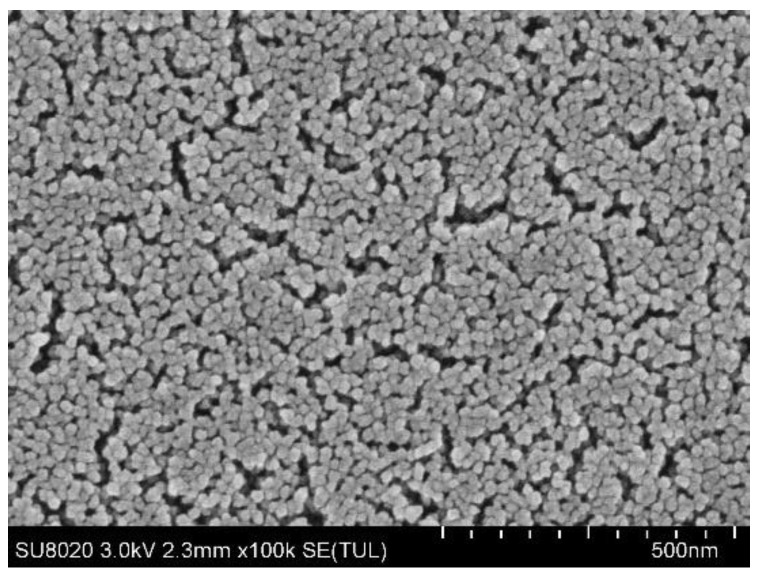
SEM image of fabricated SiO_2_ nano-particles utilizing Stober method.

**Figure 3 sensors-22-01520-f003:**
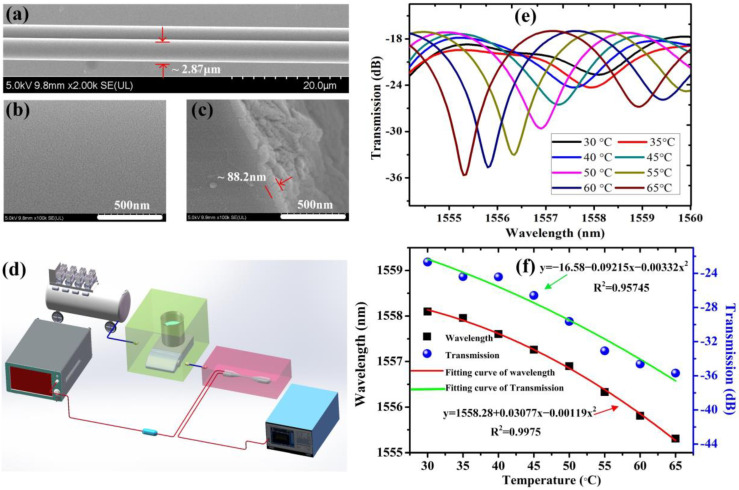
(**a**) shows a typical SEM image of uniform region of OMFs coupler; (**b**) shows microtopography of silica mesoporous coating on OMFs coupler; (**c**) shows a typical SEM image of the cross section of the fabricated sensing unit; (**d**) the schematic of detection equipment; (**e**) transmission spectra of the proposed sensor at different Ts; (**f**) response of the wavelength and intensity of the dip under different Ts.

**Figure 4 sensors-22-01520-f004:**
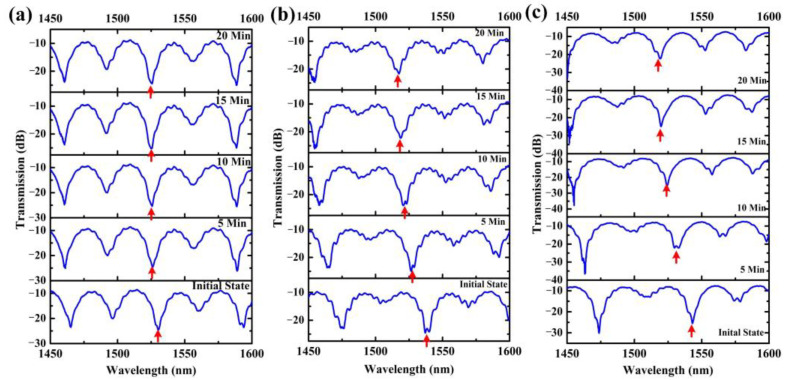
Transmission spectra evolution of the proposed sensor with 3.0 μm diameter at (**a**) 30 °C, (**b**) 50 °C and (**c**) 70 °C, the duration time of heating was 20 min in the experiment. The red arrows in the spectral responses represent corresponding dip wavelength shifts as elapsed time increases.

**Figure 5 sensors-22-01520-f005:**
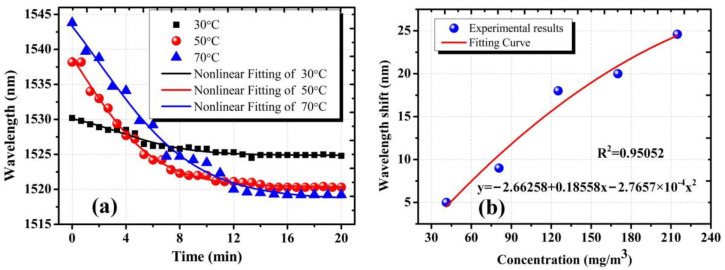
(**a**) The wavelength dips shift for the proposed sensors with 3.0 μm diameter as measurement time elapsed at 30 °C, 50 °C and 70 °C; (**b**) the relationship between wavelength shift and AMCs concentration for the as-fabrication sensors with 3.0 μm diameter.

**Figure 6 sensors-22-01520-f006:**
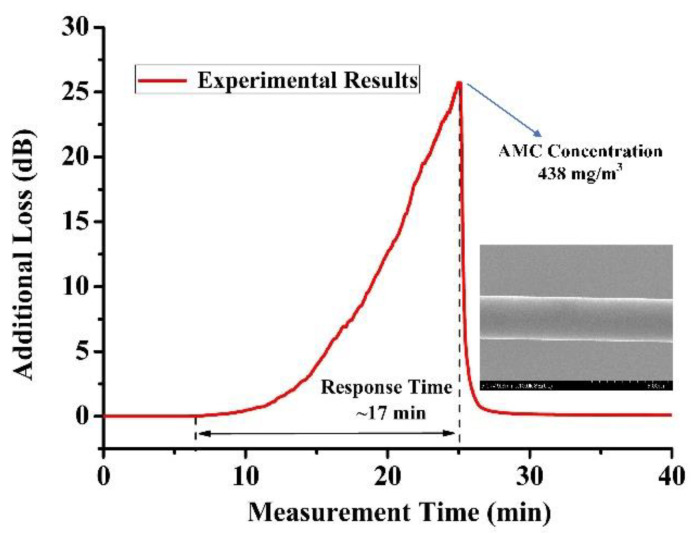
Response of the OMFs sensors at heating temperature of 98 °C, the inset shows the SEM image of the OMFs sensor with ~3.0 μm diameter.

**Figure 7 sensors-22-01520-f007:**
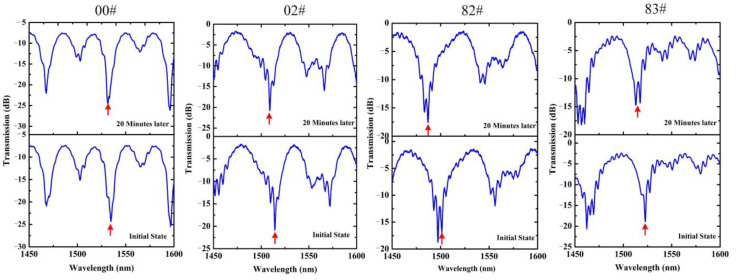
Transmission spectra evolution of grease sample 00#, 02#, 82# and 83# at coarse vacuum of ~20 Pa, and the diameter of the sensors used in the measurement process is 3.0 μm.

**Figure 8 sensors-22-01520-f008:**
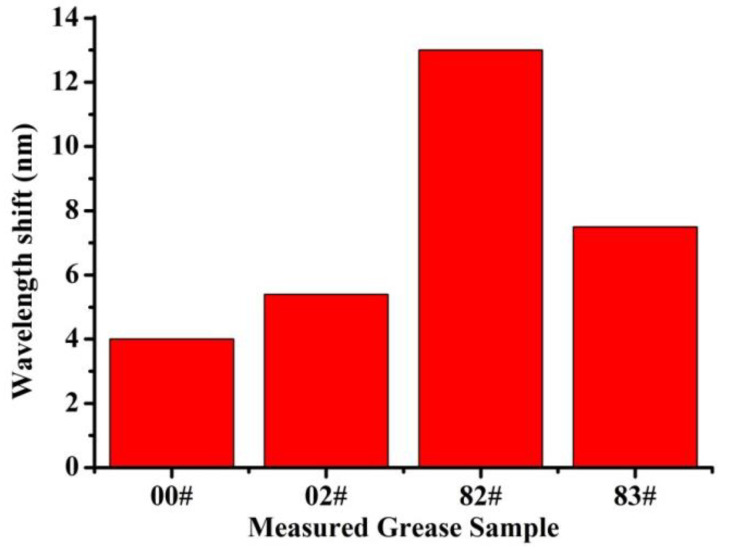
The wavelength dips shift of grease sample 00#, 02#, 82# and 83# at coarse vacuum of ~20 Pa.

## Data Availability

The datasets generated during and/or analyzed during the current study are available from the corresponding author on reasonable request.

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
