# Peer review of "A Novel Airborne Molecular Contaminants Sensor Based on Sagnac Microfiber Structure"

_sensors, 2022, doi:10.3390/s22041520_

Round 1

Reviewer 1 Report

The authors have performed a fair investigation on detecting airborne molecular contaminants using microfiber Sagnac interferometers. However, the presentation of the article results and discussion much need to improve the article quality. So, the article can be accepted after a revision by addressing the queries.

The manuscript entitled” A novel airborne molecular contaminants sensor based on Sagnac microfiber structure “by Guorui Zhou et al. reports on a fiber optic sensor based on a microfiber Sagnac interferometer for detecting airborne molecular contaminants. This research might be the continuation of the already published article entitled “Sensing of airborne molecular contaminants based on microfiber coupler with mesoporous silica coating” by the authors. The research work is inspiring, and this study of microfiber Sagnac interferometers based sensors is much needed in the fiber optics sensors regime. 

 The authors have performed quite good research on this investigation. The language used in this article is reasonable. The critical discussion element to support the results/finding, citations for the statement are missing. However, the language presented in the article, uniformity in the result is improved the article quality will be further enhanced. Overall, the article is worthy. However, all the results are needed continuity, uniformity can improve the fineness of this manuscript. I want to address a few queries on this manuscript, and it will help improve the quality of the article. Please find the comment below.

  1. Page no 1, line number 40, authors have stated, “Meanwhile, the extreme environment such as vacuum or intense laser radiation also limits or restricts the application of other detection technologies [6,7]”. The sentence is not clear along with the reference. Would you please reframe the sentence according to the reference cited? 
  2. Page no 1, line number 44. authors have stated,” sensors with thinner diameter structure have successfully achieved higher sensitivity” [*]. Would you please provide an appropriate reference if you have any?  
  3. Page no 2, line number 50, the reference cited in the sentence, Is the research work experimental one / theoretical simulated one. If it is theoretical, please provide any reference with a microfiber based evanescent field sensors. It will be suitable for people reading your article. 
  4. Page no.2, line number 52, “In OMFs sensors, there are several major types that have been developed recently, which include Mach-Zehnder OMFs interferometric sensors [10- 12]”. 

The Mach-Zehnder OMFs interferometric sensors are not just the three articles. There are plenty of reports of different MZI structure types used on fiber optic communication and sensing. The most recent work from K Ramachandran and Naveen Kumar discusses the other Mach-Zehnder OMFs interferometric sensors of different MZI types used for sensing various parameters. 

[*] Ramachandran, K. and Kumar, N., 2021. Comparative spectral tuning and fluctuation analysis of an all-fiber Mach–Zehnder interferometer and micro Mach–Zehnder interferometer. Journal of Optics23(11), p.115702.

There are many review articles that focus on the Mach-Zehnder OMFs interferometric sensors. Would you please provide some of them? 

  1. Page no.2, line number 58, Due to its truly path-matched interference mode, Sagnac interferometric sensor shows its superior temperature stability, high sensitivity, which renders it suitable, particular for AMCs sensing [*]. Would you please provide an appropriate reference if you have any
  2. Page no.2, line number 65, authors have stated that “Through verifying the proposed AMCs sensors, it indicates that they would have high sensitivity “. Would you please specify/justify how high concerning any existing schemes? 
  3. Page no.2, line number 76, authors have stated that “Secondly, although Sagnac OMFs loop and OMFs coupler have similar structures, they are completely different optical structures and have totally different characteristics.”. It is a vague statement, and please make it an appropriate sentence in terms of light propagation. 
  4. Page no.2, line number 81, authors have stated that “Simultaneously, due to its unique structure, it owns more symbolic wavelength dips, better recognizability in the measurement process, and better usability. And its technical feasibility has been carried out in work.”

 Please reframe the sentence in a better way, and it does not convey the actual meaning. 

  1. Page no. 4, line number 133, authors state, “Here, for the Sagnac OMFs loop structure, V>1. ??/ ??2< 0, which indicates that weak coupling dominates the whole coupling process and the wavelength of the dip blueshifts as the RI of the surrounding medium n2 increases.”

 It is not clear. Would you please elaborate on it concerning Eqn 3?  If the V number is just greater than 1, how many modes will exist on the coupling region? 

  1. Page no. 4 line number 147, author’s states “As shown in the SEM image, the shapes of silica nano-particles with a diameter of 20.0nm are not very regular, and the mesoporous coating formed on the glass slide substrate by dense packing and or a van der Waals attraction [*]. And porosity increases due to abundant voids of coatings[*], Please provide an appropriate reference if you have any
  2. Page no. 6 line number 216, author’s states s shown in 216 Figure 3(e), as the increased from 30℃ to 65℃, the wavelength dip exhibited a redshift 217 and a decrease in intensity due to the combined effects of the thermo-optics and the thermal expansion in the tapered waist region”. Would you please justify it?
  3. Page no. 6, line number 240, Figure 4, Have you observed the spectral stability/repeatability at 30oc?
  4. Page no. 6, line number 240, Figure 4, why the second notch/dip ~1500nm is suppressed when the temperature increases? 
  5. Page no. 7, line number 265, figure 5(a) the figure caption temperature is different from the temperature mentioned inside the graph? 
  6. Page no. 7 line number 265, figure 5(a) the wavelength dip is starting from 1545 nm say, but in figure 3 (e ) the wavelength dip is 1555 nm above? Why the discrepancy?
  7. Page no. 7 line number 265” Transmission spectra comparison of grease sample 00#, 02#, 82# and 83# before and after evaluation” what these numbers “00#, 02#, 82# and 83#” corresponding to? How is it different from each other? 
  8. Page no. 8, figure 6, the spectra have a lot of fluctuation/distortion. Why? 
  9. Page no. 8, figure 6, Is the wavelength notch/dip stable, or is the spectrum moving? If then how accurately evaluate the wavelength shift? 
  10. The wavelength shift-based sensors are available, so authors are specifically interested in the Sagnac interferometer. 
  11. How about the durability and packaging of this sensor? 
  12. The theoretical work mentioned is already published in article31. What authors contributed to the present article? As the authors state on page no. 2, line 78, “we have carried out theoretical analysis and formular derivation again, rather than numerical analysis.”. Would you please justify it?  

Reviewer 2 Report

Authors presented an Airborne Molecular Contaminants (AMCs) sensor based on a Sagnac interferometer (SI). Moreover, they described the fabrication procedure of the SI and explained the spectral behavior of the sensor as a function of the temperature and the AMC concentration. I consider that the work is interesting but I would like to suggest that authors must perform major corrections before the manuscript can be considered for publication. Some comments and suggestion are as follows:

  • In figures 4 it can be appreciated that with temperature not just the deep position of certain fringes changes, but also the overall spectrum is modified. For instance in the spectrum recorded at 70ºC some peaks are considerably attenuated. Please discuss this issue in the manuscript and explain of this can introduce some effects in the measurement range.
  • The inset table of figure 5a it is illegible, I would like to suggest to present it as a table of the manuscript if it important for the article.
  • In figure 5b, it response is considering a fixed temperature?, please indicate the experimental conditions.
  • There are some important aspects related to the technical characteristics of the sensing system that need to be defined in the manuscript. For instance what is the measurement dynamic range of the sensor?, what is the minimum and the maximum concentrations that can be detected with the sensor?, what is the resolution and the sensibility of the sensing system?.
  • Also discuss the limitation in the measurement dynamic range due to the periodicity of the spectral fringes. Since, one fringe will shift practically over 1 free spectral range and after it will overlap the spectral region of the next fringe. So the wavelength position shifting of a fringe will have a behavior similar to a sawtooth waveform. It issue can be observed in figure 3e of the manuscript. It issue is presented in many interferometric sensors and some authors have explained this type of behavior and their relationship with the dynamic measurement range. For instance I suggest to consider the following reference: "Tunable Optical Filter Based on Two Thermal Sensitive Layers," IEEE Photonics Technology Letters, Vol. 30, No. 20, 2018.
  • Please discuss in a detailed way the cross-sensitivity effect of the sensor, since it is sensitive to temperature and to the concentration of the AMC. Also discuss in practical situation can be minimized this effect.
  • Please explain the term as-fabricated, since is a bit confusing the word so please define it or replace with a more standard terminology.

Round 2

Reviewer 2 Report

I have carefully checked the author’s reply to my comments. I have noted that authors confirmed that accepted the comments but basically I have not found that they provided a clear answer to each point. Even some comments about that tables embedded in tables are illegible authors commented that were attended but still basically illegible in the revised version. Moreover, there are some technical points that are quite confusing in the revised version, for instance in figure 5b it is presented the response of the sensor to the AMC concentration, but in the reply letter to question 3, they also mentioned that the temperature was not fixed that “In the experiment, the heating temperature varied from 30℃ to 70℃ at intervals of 10℃. In order to achieve equilibrium between adsorption and desorption of AMCs, the duration time of heating was set at 20 minutes, and the spectral response at each temperature was recorded.” Here based on the mentioned by authors the curve presented in figure 5b is the response of the sensor to both the AMC concentration and the temperature. Therefore, as far as can be understood the sensor response depends simultaneously to both variables and therefore the paper must be analyzed and addressed in a different way. Based on these comments I consider that authors must work in the detailed description of the principle of operation of the sensor in order to enhance the manuscript. Also a detailed characterization of the sensor response to temperature, to the AMC concentration and the two variables must be provided, in order to understand how the sensor will behave when these parameters are varied. Also I would like to suggest to authors to take into account the comments provided by the reviewers. Finally I would like to suggest to review the English edition of the paper since there are some typos (ie in figure 5a) that must be corrected.
